# Regulatory miPEP Open Reading Frames Contained in the Primary Transcripts of microRNAs

**DOI:** 10.3390/ijms24032114

**Published:** 2023-01-20

**Authors:** Tatiana N. Erokhina, Dmitriy Y. Ryazantsev, Sergey K. Zavriev, Sergey Y. Morozov

**Affiliations:** 1Shemyakin-Ovchinnikov Institute of Bioorganic Chemistry, Russian Academy of Sciences, 117997 Moscow, Russia; 2Belozersky Institute of Physico-Chemical Biology and Biological Faculty, Lomonosov Moscow State University, 119991 Moscow, Russia

**Keywords:** microRNA, microRNA primary transcripts, translation of microRNA primary transcripts, short open reading frame, micropeptide, miPEP, transcription of pri-miRNA

## Abstract

This review aims to consider retrospectively the available data on the coding properties of pri-microRNAs and the regulatory functions of their open reading frames (ORFs) and the encoded peptides (miPEPs). Studies identifying miPEPs and analyzing the fine molecular mechanisms of their functional activities are reviewed together with a brief description of the methods to identify pri-miRNA ORFs and the encoded protein products. Generally, miPEPs have been identified in many plant species of several families and in a few animal species. Importantly, molecular mechanisms of the miPEP action are often quite different between flowering plants and metazoan species. Requirement for the additional studies in these directions is highlighted by alternative findings concerning negative or positive regulation of pri-miRNA/miRNA expression by miPEPs in plants and animals. Additionally, the question of how miPEPs are distributed in non-flowering plant taxa is very important for understanding the evolutionary origin of such micropeptides. Evidently, further extensive studies are needed to explore the functions of miPEPs and the corresponding ORFs and to understand the full set of their roles in eukaryotic organisms. Thus, we address the most recent integrative views of different genomic, physiological, and molecular aspects concerning the expression of miPEPs and their possible fine functions.

## 1. Introduction

Regulatory microRNAs (miRNAs) are short double-stranded molecules derived from rather long precursors called primary miRNAs (pri-miRNAs) and transcribed in the nucleus from chromosomal DNA by RNA polymerase II. These pri-miRNAs have been shown to contain cap-structure and poly(A)-tail at their ends and include internal imperfect hairpin structures, which are cleaved by ribonuclease complex (Figure 1). The stem-loop structures of plant pri-miRNAs are variable in length (from 60 nt to over 500 nt). This process occurs in distinct sub-nuclear bodies, namely, dicing bodies (D-bodies). Proteins DCL1, Hyponastic Leaves 1 (HYL1), and Serrate (SE) are the key core components of a processor complex located in D-bodies. DCL1, an RNAse III-type endoribonuclease, is responsible for the cleavage of pri-/pre-miRNAs [1,2,3,4]. Usually, the first DCL-dependent step is the formation of pre-miRNA when the hairpin structure is processed from the loop-distal site (Figure 1). The second processing step is cutting off the loop. Resulting in short imperfect double-stranded RNAs with the 3′-extending two-nucleotide overhangs undergoing subsequent 3′-terminal methylation by the RNA methyltransferase HEN1. This leads to the formation of the nucleus-localized mature mi-RNAs [1,2,3,4]. Methylated RNA duplexes are known to be transported from the nucleus by the protein Hasty (HST). After transport to the cytoplasm, plant mature miRNAs interact with a protein complex containing the Argonaute ribonuclease (AGO1), which selects one of the strands of the double-stranded miRNA. The other strand (the passenger strand, miRNA^∗^) is degraded. Complementary partial base-pairing of the guide strand with its target mRNA either leads to precise cleavage of this template or suppresses its translation (Figure 1). AGO proteins contain conserved PAZ, MID, and PIWI domains. The MID and PAZ domains bind to the 5′ phosphate and 3′ end of small RNAs, respectively, while the PIWI domain cuts target RNAs through its endonuclease activity. This process plays an essential role in the regulation of the absorption of nutrients and plant development, namely, root initiation, leaf development, vascular development, flower development, phase transition, and seed development [1,2,3,4]. In addition, plant and animal miRNAs are regarded as important stress-responsive gene-regulatory factors [5,6].

It has been previously accepted that the pri-miRNAs represent non-protein coding RNAs [7,8], and its RNA segments upstream and downstream of the hairpin, which corresponds to pre-miRNA, degrade rapidly after excision. Indeed, it was found that pri-miRNA molecules have a very low abundance in plants, contrasting with relatively high levels of mature miRNAs [9]. Interestingly, heat stress-induced alternative intron splicing that keeps the pre-miR400 stem-loop in the translated cytoplasmic At1g32583 gene mRNA of Arabidopsis plants may prevent pre-miR400 processing and, thus, reduce miR400 accumulation [10].

The rapid increase in the use of bioinformatics and integrative “omics” approaches for the analysis of gene expression processes in plant and animal cells have allowed massive finding the translatable short open reading frames (ORFs) within the numerous pri-miRNAs and other long “non-coding” RNAs [11,12,13]. For example, translatomics experiments were used as an experimental method to assess whether a particular transcript is associated with polysomes. Particularly, the translating ribosome affinity immunopurification (TRAP), or ribosome profiling method, has been developed to measure ribosome-protected RNA fragments [14,15]. This method has revealed that the Arabidopsis plant pri-miRNA fragments can be associated with ribosomes and, thus, contain translating open reading frames (ORFs) [12,16].

In the past years, the presence of short-translated ORFs in the primary transcripts of plant miRNA and the identification of the encoded peptides (miPEPs) have been comprehensively reviewed [13,17,18,19,20,21,22,23,24,25,26]. In general, the use of integrative “omics” methods has allowed the fine studies of the molecular processes that plant cells. In the literature, many review papers [13,19,20,21,22] have considered “omics” methods in an extensive way. However, in our review, we address mostly the retrospective and the most recent integrative views of different genomic, physiological, and molecular aspects concerning the expression of miPEPs and their possible fine functions.

## 2. Retrospective of miPEP Finding and Analysis in Plants

In a pioneering study [27], it has been shown that primary transcripts of many *Arabidopsis thaliana* microRNAs (pri-miRNAs) contain short ORFs in the 5′-proximal regions. These miPEPs were found to be from 3 to 59 amino acids long and showed no significant similarity, suggesting that each miPEP is specific for its corresponding miRNA. Evidence obtained by in vivo overexpression of the corresponding miPEP ORFs or external spraying of plants with synthetic peptides shows that miPEP (*At*-miPEP165a, 18 aa long) from *Arabidopsis* (family *Brassicaceae*) is able to activate the transcription of its own pri-miRNA messenger. Moreover, root modifications, namely, stimulation of main root growth and decreased lateral root formation, have been found after external application of *At*-miPEP165a when the miPEP enters into *Arabidopsis* roots by both passive diffusion and endocytosis-associated processes [28]. Evidence for the expression of miPEP165a *in planta* was supported by a combination of in vivo studies of artificial translational GUS fusions and immunolocalization studies. Similar results have been observed in studies of miPEP171b (20 aa long) from barrel clover (*Medicago truncatula*, family *Fabaceae*), where synthetic miPEP increases endogenous expression of *Mt*-miR171b and leads to a decrease in the density of the lateral roots [27,29]. Importantly, external plant treatments with *Mt*-miPEP171b and overexpression of this miPEP resulted in stimulation of the mycorrhization process. Moreover, the use of synthetic miPEPs encoded by miR171b homolog genes in plants *Lotus japonicus* (family *Fabaceae*), *Solanum lycopersicum* (family *Solanaceae*), and *Oryza sativa* (family Poaceae) for watering these plants also lead to increased miR171b expression and mycorrhization, despite that four compared miPEP171b species were sequence unrelated [30]. Later, *Gm*-miPEP172c has been shown to control nodulation in soybean (*Glycine max*, family *Fabaceae*) [31]. Treating soybean plants with a solution containing a synthetic peptide *Gm*-miPEP172c led to an increased number of nodules. This enhanced nodule formation is also correlated with increased pri-miR172c and *Gm*-miR172c expression [31].

Further studies have confirmed the role of miPEPs in the pri-miRNA transcription regulation of many other species of miRNA in different plants. Particularly, to identify miPEP encoded by pri-miR858a, the putative open reading frames (ORFs) were screened in 1000 bp region upstream from pri-miR858a in *Arabidopsis thaliana*. Transient expression of the in-frame fusions of β-glucoronidase (GUS) reporter gene with several translation initiation codons from the pri-miRNA 5′-proximal area and histochemical GUS staining suggests the existence of 135 bp long ORF encoding small peptide (miPEP858a) of 44 amino acids [32]. CRISPR-based miPEP858a-edited and miPEP858a overexpressing plant lines show altered plant development phenotypes. These phenotypes include reduced plant growth, delayed flowering, and the enhanced accumulation of flavonoids, anthocyanin, and reduction in the level of lignin accumulation [32]. Generally, the importance of cross-talk between miPEP858a/miR858a and phytosulfokine (PSK4 gene) in regulating plant growth and development in *Arabidopsis*, including auxin responses, has been comprehensively studied [33].

The search of miPEPs in genera *Arabidopsis* and *Brassica* has revealed additional plant miPEP, *At*-miPEP156a (33 amino acids long), which is evolutionarily conserved in many plants of the family *Brassicaceae* [34,35,36]. These conserved miPEPs are able to change plant phenotype upon exogenous application to plant seedlings; particularly, they have a positive effect on the growth of primary roots. Application of the synthetic peptides shows that miPEP156a may activate the synthesis of the pri-miR156a [35]. Moreover, the ability of the synthetic miPEP to transport into the nucleus and interact with nucleic acids has been revealed [34]. Strikingly, evolutionarily conservation of miPEPs among plants of the family *Brassicaceae* was observed only for two more miRNA genes except for miPEP156a, namely, *At*-miPEP 165a and *At*-miPEP164a [12].

Short ORFs encoding miPEPs have been also found in grapevine plants (*Vitis vinifera*, family *Vitaceae*). First, it was published that *Vvi*-miPEP171d may act as a regulator of root growth. Exogenous treatment with miPEP171d1 has shown that the transcription level of the corresponding miRNA and the number of adventitious roots was increased in the grape tissue culture plantlets [37]. Second, the exogenous application of *Vvi*-miPEP164c to suspension-cultured grape berry cells also enhanced the transcription of the corresponding miRNA. This leads to a more pronounced miRNA-dependent silencing of the grapevine transcription factor *Vv*MYBPA1 and subsequently to significant inhibition of the proanthocyanidin biosynthetic pathway [38]. In the last three years, the list of bioinformatically predicted miPEPs is increased constantly. Particularly, several novel miPEPs of peanut plants *Arachis hypogaea* [39] and additional peptides conserved between the other four species of the family *Fabaceae* have been revealed [40]. Moreover, two novel small ORFs of maize encoding *Zma*-miPEP159d and *Zma*-miPEP2275d have been found using bioinformatics, ribosome profiling, and mass spectrometry [41].

## 3. Potential miPEPs in Mosses

The plant miPEPs revealed so far have been found only in flowering plants (dicotyledons and monocotyledons) (see above). However, the question of the possible encoding of miPEPs in other eukaryotic taxa is very important for understanding the evolutionary origin of such peptides [11]. In this respect, it is important that mosses have been shown to encode many peptides in non-coding RNAs, which functions are mostly obscure [42]. Recently, we have revealed the NCBI annotated small predicted protein (PHYPA_019725) from moss *Physcomitrella patens* (accession PNR39447, 122 aa in length) that is encoded in chromosome 15 by the 5′-proximal region of the *Ppt*-miR160a gene (https://mirbase.org/summary.shtml?fam=MIPF0000032 accessed on 26 September 2022) [43]. A moderately similar protein was found to be encoded in *P. patens* chromosome 9 by the 5′-proximal region of *Ppt*-miR160f gene (https://mirbase.org/summary.shtml?fam=MIPF0000032 accessed on 26 September 2022). Both proteins contain highly similar sequence block in the middle area (Figure 2A). Significantly similar amino acid sequence blocks were detected in short proteins encoded by the 5′-terminal regions of pri-miR160 in three more mosses (*Pohlia nutans*, *Ceratodon purpureus,* and *Syntrichia caninervis*), for which partial genome sequences have been determined to date (NCBI accessions JAKGBK010000017, JACMSB010000170, and JADDRJ010000003, respectively) (Figure 2B). It is important that additional TBLASTn analysis revealed coding of the highly conserved miPEP160a peptide block in transcriptomes and genomic fragments of a dozen of Bryopsida mosses.

Interestingly, very similar peptide blocks are found in short proteins encoded by transcriptomes of mosses from the classes *Polytrichopsida* and *Takakiopsida* (Figure 2B). Considering that the latter moss class is the most ancient phylogenetic branch in the Bryophyta division (https://phytozome-next.jgi.doe.gov/ accessed on 26 September 2022), this strongly suggests a high evolutionary conservation of the protein analogs of *Ppt*-miPEP160a among all mosses.

## 4. Fine Molecular Mechanisms Underlying Ability of Plant miPEPs to Activate pri-miRNA Transcription

In the past years, some important questions have been raised in relation to the fine molecular mechanisms underlying plant miPEPs functions. Particularly, how sequences of the miRNA genes and/or pri-miRNAs are recognized by miPEPs to activate pri-miRNA transcription, and if this recognition is really taking place, how the transportation of exogenic and endogenic miPEPs into the nucleus occurs (Figure 1)? The last question has only phenomenological answers up to date. Indeed, transportation of endogenic, plant-expressed *Mt*-miPEP171b efficiently occurs into small nuclear bodies [44], and the exogenic miPEP165a of *Brassica* species actively migrates into nuclei of phloem and leaf cells [35].

However, a paper published in the last year [12] has reported new views on the mechanisms of the pri-miRNA transcription activation by the homologous miPEPs. Particularly, it was detected that up-regulation of pri-miR156a in all tested *Brassicaceae* plants treated with the heterologous miPEP156a peptides from different species still works if there were only 10–15% mismatches. However, if there are more than 70% mismatches, as in the case of the *Brassicaceae* miPEP167a, such heterologous up-regulation is not observed [12]. The experimental data have revealed that miPEP specificity in the miRNA transcription activation may rely on a direct physical interaction between the miPEP and its own coding ORF located in the nascent sequence of the transcribing pri-miRNA [12]. Moreover, it was suggested that the miPEP-interacting RNA region should have only a specific linear set of codons, but not a strongly specific nucleotide sequence, to drive positive transcription activation in response to a particular miPEP. Anyway, assuming a large number of tested miPEPs (9–33 aa in length) with unrelated sequences, it can be hypothesized that almost any peptide with the size up of to three-four dozen amino acids may activate transcription of its own RNA template. All these data might explain why there is no selection pressure on sequences of plant miPEPs [12]. The molecular mechanism for such an unusual phenomenon could be explained by miPEP-induced modulating two stages of pri-miRNA transcription, namely, initiation (enhanced frequency of transcription starts) and/or elongation (enhanced speed of RNA polymerase progressing along template DNA) (Figure 1).

A recent study of the plant pri-miRNA transcription and processing has shed a light on the miPEP transcription activation ability. It was found that pri-miRNA processing may occur co-transcriptionally and may start from the loop of the pre-miRNA hairpin. Moreover, R-loops, frequently formed near the transcription start site regions of miRNA genes, influence co-transcriptional pri-miRNA processing. Importantly, co- and post-transcriptional miRNA processing events may co-exist for most miRNAs in a dynamic balance. In general, the R-loops, positioned around the transcription start site region of miRNA genes, promote co-transcriptional pri-miRNA processing. However, less efficient co-transcriptional processing still occurs in miRNA loci forming no R-loop [45]. It is known that during transcription, the nascent RNA strand can base pair with its template DNA and may displace the non-template strand as R-loop of ssDNA. While short RNA-DNA hybrids may form transiently during the normal transcription process, R-loops are longer in comparison with the former structures and occupy 100–2000 bases [46]. Importantly, a change in the transcription elongation rate may affect the pri-miRNA folding and processing, and R-loop formation could potentially hamper efficient pri-miRNA transcription. Particularly, R-loops may repress the initiation of transcription by blocking transcription factor binding at promoters [46]. In general, the discovery that the formation of R-loops near the transcriptional start site of miRNA genes promotes co-transcriptional processing of pri-miRNA, provides a novel regulatory scenario re-defining the difference in the functioning of co-transcriptionally and post-transcriptionally matured miRNA. Thus, the identification of proteins, such as RNA-helicases, which help in resolving these R-loops, is imperative to study the peculiarities of the expression and potential functions of miRNAs [45]. We propose that miPEPs may act as helicase-like proteins because of their ability to bind their own ORFs in the 5′-terminal pri-miRNA regions potentially positioned in the R-loop near transcription start. Seemingly, this binding enhances the melting of RNA-DNA hybrids. Indeed, miPEPs are found to activate pri-miRNA transcription (see above), and, specifically, *At*-pri-miRNA156a producing conserved miPEP [24] is known to be mostly processed at the post-transcriptional step of biogenesis.

## 5. Potential miPEPs in Metazoa

In recent years, evidence was obtained that pri-miRNAs are also translated in metazoan cells [11]. Particularly, the human gene of pri-miR-22 can produce miPEP induced in response to viral infection. However, its function is still unknown [47]. It was also shown that pri-miR-200 can be translated to form two miPEPs, which inhibit the migration of prostate cancer cells and regulate the synthesis of Vimentin, the target of the MIR200 gene [48]. In addition, it was found that human miPEP155 (17 aa in length) controls the presentation of antigens mediated by a class II histocompatibility complex [49]. Peptide miPEP133 (133 aa long) encoded by pri-miR34a has been identified as a tumorogenesis suppressor localized in mitochondria and represents also a positive regulator of pri-miR-34a/miR-34a expression [50]. A more recent paper has described a peptide called miPEP31, which is encoded by pri-miRNA-31 [51]. This peptide is efficiently expressed in regulatory T cells and promotes their differentiation. The results obtained show that miPEP31 suppresses miR-31 expression and dramatically inhibits experimental autoimmune encephalomyelitis. It is assumed that miPEP31 acts as a transcriptional repressor inhibiting the miRNA-31 expression and is involved in maintaining immune homeostasis by stimulating the differentiation of regulatory T cells [51]. Thus, this peptide may act in a molecular way drastically different from plant miPEPs and can be considered as a DNA-binding repressor protein interacting with the pri-miRNA gene promoter. It may represent a potential therapeutic peptide for the modulation of microRNA expression and the treatment of autoimmune diseases [51]. Interestingly, it was shown recently that overexpression of human micropeptides miPEP155, miPEP497, and miPEP200a has no influence on the synthesis of their own pri-miRNA/miRNAs in contrast to miPEP31 and miPEP133 [52]. This suggests that the feedback positive regulation observed with plant miPEPs (see above) is not a general rule for human miPEPs.

It is important that the insect *Drosophila melanogaster* has also been found to produce miPEPs. In particular, a small ORF was found in pri-miR-8 encoding a potential peptide with a length of 71 amino acid residues, which was named miPEP-8. The expression of this peptide affects the development and survival of flies, but miPEP-8 expression has no effect on pri-miR-8 transcription [53]. Interestingly, although *At*-miPEP-165a acts as a positive regulator of *At-miR165a* gene expression in heterologous plant *N. benthamiana*, it shows no upregulation but rather, a downregulation of *pri-miR-165a* in *Drosophila* S2 cells [53].

In the current year, the principally important findings have been made with *Drosophila* miPEP-encoding ORFs from pri-miR-8 and pri-miR-14 [54]. Strikingly, the accumulation of these pri-miRNAs is sensitive to the translation-inhibiting drug cycloheximide, suggesting the involvement of translational events in the regulation mechanisms. Moreover, this regulation is seemingly independent of the nucleotide sequences of the miPEP ORFs but rather relies upon the presence or absence of the similarly positioned ORFs in the 5′-terminal regions of pri-miRNA. A similar mechanism of the negative miPEP-ORF-dependent translation regulation has also been revealed for human pri-miR497 [54]. Thus, in Metazoa, mechanisms of miPEP- and miPEP-ORF-dependent pri-miRNA regulation are found to be subject to significant variations.

## 6. Conclusions and Future Directions

Currently, it is obvious that in animals and plants, primary microRNA transcripts (pri-miRNAs) can be translated, similarly to conventional mRNAs, to form peptides (miPEPs). These peptides are involved in regulatory pathways and have a size from several to several dozen residues. Assuming the above-mentioned data on the coding properties of pri-miRNAs, one formal question remains unclear, namely, can we still consider these transcripts as conventional long non-coding RNAs [8]? The most fundamental criteria used to distinguish long ncRNAs from mRNAs represent the ORF length and position. Evidently, short putative ORFs can be expected to occur by chance within long noncoding sequences. The investigations on ncRNAs originally used a cutoff of 300 nt (100 codons) in order to discriminate between putative mRNAs and ncRNAs, as it was applied to the soybean *ENOD40* lncRNA encoding in the 5′-terminal part two small functional peptides of 12 and 24 amino acids and playing also a specific role independent of peptide coding, namely, interaction with the plant RNA BINDING PROTEIN 1 and SMALL NODULIN ACIDIC RNA-BINDING PROTEIN [13,55]. Currently, obtaining novel information on the functional potential of the long ncRNAs has resulted in a new classification scheme where RNAs with both protein-coding and noncoding functions are referred to as bifunctional RNAs, or cncRNAs (coding and noncoding RNAs) [53,56,57].

Accumulation of the new data concerning the mode of action of miPEPs in plant and animal organisms resulted in quite a mosaic picture. Generally, the potential fine molecular mechanisms of the miPEP and/or miPEP ORF activity can be classified as follows: transcriptional regulation (Figure 1) and translational regulation of pri-miRNA. The first mode of action can be conventional or non-conventional [35,51]. In principle, conventional regulation of the pri-miRNA transcription by miPEPs may include positive [35] or negative [51] influencing the initiation step, which is known to represent a complex process involving dozens of protein factors [58]. Particularly, in plants, the Mediator protein complex plays a crucial role in attracting Pol II to the promoters of microRNA genes. This complex includes MEDIATOR 20A (MED20A), MEDIATOR 17 (MED17), and MEDIATOR 18 (MED18) [58]. In the corresponding mutants, the level of pri-miRNA transcription is greatly reduced. Some other transcription factors, such as the nuclear proteins RBV (nuclear WD40 domain-containing factor), CDC5, and DOF, also cause microRNA transcription modulation. These DNA-binding proteins positively regulate the attraction of Pol II to microRNA promoters and their activity [58]. Other protein factors, such as SWINGER (SWN), CURLY LEAF (CLF), CHR2, and PICKLE (PKL), additionally regulate pri-miRNA transcription due to their activity in chromatin remodeling in the promoter region of some pri-miRNA genes, particularly, miR156a gene [58]. We have previously suggested that miPEP-156a is conserved in *Brassicaceae* plants and is able to interact with chromatin in the promoter region and, thus, activate the synthesis of pri-miR156a [35]. Involvement of the evolutionary conserved *At*-miPEP164a [12] in positive regulation of transcription initiation by the attraction of Pol II can be also proposed.

However, transcription activation of miRNA genes by these and other plant miPEPs seems to represent a non-conventional process involving the interaction of the peptides with their own coding ORFs located in the nascent chain of transcribing pri-miRNA [12]. Principle novelty of this process concerns previously unreported mechanisms when peptide molecule specifically binds RNA region having only a specific linear set of cognate codons. Thus, miPEPs having any amino acid sequence should interact with RNA ORFs encoding them. Future directions in studies of this remarkable phenomenon should, first of all, include confirmation of the previously published data [12]. The next point to be clarified in connection to this phenomenon concerns the following question: how can miPEPs interacting with nascent chains of transcribing pri-miRNAs increase the efficiency of transcription? Our hypothesis (see above) that miPEPs may act as helicase-like proteins binding to their own ORFs in the 5′-terminal pri-miRNA regions, and thus, significantly eliminating R-loop formation near transcription start, should also be verified. Interestingly, mutations of RH27, a nucleus-localized DEAD-box RNA helicase, which associates with pri-miRNAs, result in the inhibition of the accumulation of some miRNAs and their precursor transcripts in shoot apices and root tips of Arabidopsis [59].

As indicated above, some animal miRNA genes show the involvement of translation-dependent events in the regulation of accumulation and processing of pri-miRNA [54]. It was shown that this regulation is probably based on the presence or absence of the similarly positioned ORFs in the 5′-terminal region of pri-miRNA. Particularly, miPEP ORFs of *Human* pri-miR-497, as well as *Drosophila* pri-miR-8 and pri-miR-14, negatively regulate pri-miRNA accumulation [54]. At first glance, this effect can be connected with Nonsense-mediated mRNA decay (NMD), which was initially described as a quality control mechanism to remove transcripts harboring a premature termination codon and, thus, having an extremely long 3′-untranslated region. Importantly, short reading frames in the mRNA 5′-UTRs can also induce NMD. Proteins UPF1 (RNA helicase bound to 3′-UTRs), UPF2, and UPF3 represent core NMD factors selecting mRNA molecules with aberrant translation termination [60]. Previously, it has been shown that many human miRNA genes produce the NMD-sensitive transcripts, and UPF1 knockdown results in increased RNA levels [61]. However, the NMD inhibitor of the UPF1-dependent pathway has no effect on pri-miRNA levels in the case of *Human* pri-miR-497 [54]. Thus, further studies are obviously required to fully understand the phenomenon connected with the miPEP ORF-regulated translation-dependent accumulation of some animal pri-miRNAs.

To summarize, we should stress that the field connected with molecular and bioinformatics studies of miPEP ORFs in plant and animal pri-miRNAs has made tremendous progress in the last seven years. In this review, we have discussed how, with the acquisition of newly revealed miPEPs in plant and animal organisms, the conception has evolved from a simpler autoregulatory feedback pathway into a rather complex scheme of different regulatory networks depending on the concrete organism and miRNA gene. Evidently, many fundamental questions remain about miPEP ORF prevalence and function in different eukaryotic taxons and how this field of molecular sciences may contribute to possible future biotechnological outcomes. Indeed, some recent miPEP-related papers stress miPEP application as a suitable alternative to the use of chemicals in agronomy [25,36,62,63,64].

## Figures and Tables

**Figure 1 ijms-24-02114-f001:**
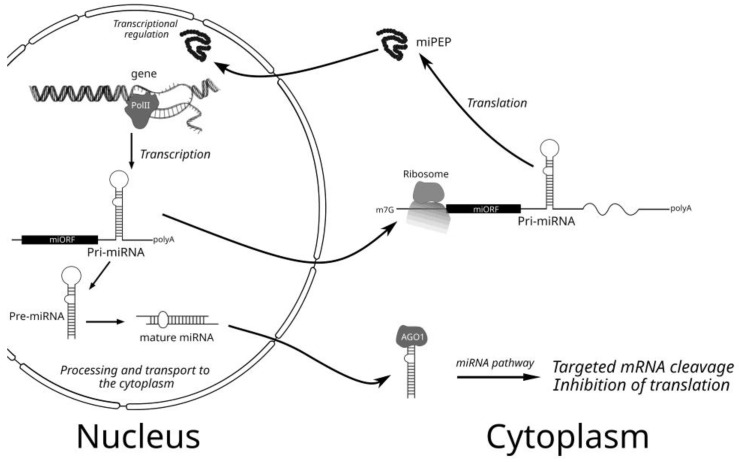
Schematic presentation of the plant cell processes of pri-miRNA/miRNA synthesis, processing, and action as well as producing and action of miPEPs.

**Figure 2 ijms-24-02114-f002:**
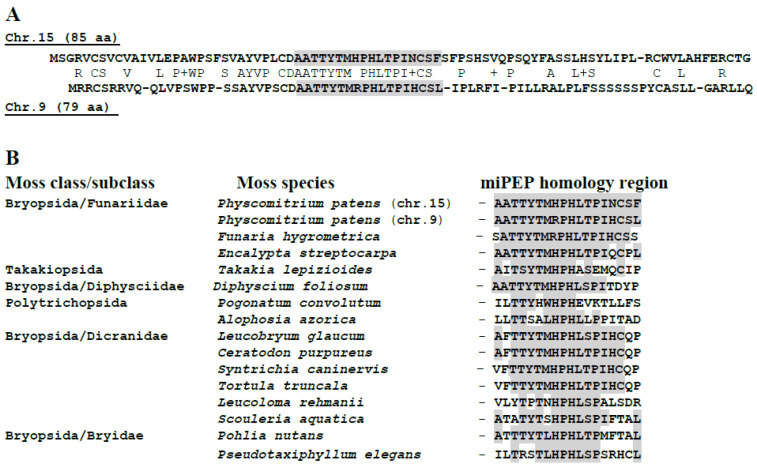
(**A**) Comparison of the C-terminal amino acid sequences of short proteins encoded by pri-miR160a genes from chromosomes 9 and 15 of *P. patens*. The sizes of proteins are indicated in parentheses. Peptide sequences of both miPEPs encoded by chromosomes 9 and 15 of *P. patens* and preserved in other mosses are marked by shading. (**B**) Comparison of amino acid sequences of highly conserved peptide blocks in three classes of mosses. Residues identical to those identified in the sequences of short *Ppt*-miPEP160a proteins are marked by shading.

## Data Availability

Not applicable.

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
