# Peer review of "Regulatory miPEP Open Reading Frames Contained in the Primary Transcripts of microRNAs"

_ijms, 2023, doi:10.3390/ijms24032114_

Round 1
Reviewer 1 Report
This paper is interesting and throws light on the regulatory functions of their Open Reading Frames (ORFs) and the en- 11 coded peptides (miPEPs).
Some comments-
-why double inverted comma in…. .. “cap”-structure and poly(A)-tail
- Many abbreviations have been used, kindly check whether every such word is initially in full form
-Kindly add miRNA role in stress citing recent works
-refer to fig 2, the homology seq ‘AA….CSF’ and ‘CSL’ is mentioned, what is the difference and why
-Add couple of lines about R-loops
-How this research will be applied in crops and agriculture
Overall a nice manifestation and authors must edit as per above
Author Response
Comments and Suggestions for Authors
-why double inverted comma in…. .. “cap”-structure and poly(A)-tail
Double inverted comma is removed (Section 1, para 1, line 4).
- Many abbreviations have been used, kindly check whether every such word is initially in full form
In a revised version, all abbreviations are used initially in full form.
-Kindly add miRNA role in stress citing recent works
Very recent references on the role of miRNA in stress regulation are added (Section 1, last sentence of para 1).
-refer to fig 2, the homology seq ‘AA….CSF’ and ‘CSL’ is mentioned, what is the difference and why
Peptide sequences of both miPEPs encoded by chromosomes 9 and 15 of P. patens and preserved in other mosses are marked by shading (see figure 2 legend).
-Add couple of lines about R-loops
Two additional sentences about R-loops are included (see section 4, para3, lines 5-8).
-How this research will be applied in crops and agriculture
The corresponding phrase and references have been added (section 6, para5, last sentence).
Reviewer 2 Report
The review submitted by Erokhina et al aims to focus on miPEP and the underlying mechanisms both in plant and in animals. To date, there no review is published on this topic and the discussion is interesting making this review of interest.
However, I would suggest to change the title of the review to that of “Genome-wide versus targeted studies of the regulatory miPEPs etc…” . The author’s title implies that the present review will cover Omics studies and how they impacted our knowledge of miPEP’s biology. Indeed, the majority of miPEP were identified by low throughput analyses and there are currently very few omics studies relating to miPEPs, if any. Although these statements remain true for short open reading frames encoded polypeptides (SEPs) in general, despite being exaggerated (currently proteomics studies do not reveal “huge” masses of small uncharacterized proteins, considerable efforts are needed to identify some), examples of applications of these approaches to miPEPs remain very sparse. For example, no miPEP were easily found by MS studies in plants (PMID 32445888), only one was identified by untargeted analyses in Drosophila (PMID 35721519) and few in humans (PMID 33810468). For me, and to the best of my knowledge, only 3 endogenous miPEPs were clearly identified by mass spectrometry so far (PMID: 35721519, 32671205 and 35343645, in the last two immunoprecipitations of the miPEPs were needed for their detection), all in animals.
In addition, certain statements of the authors can be misleading. For example, the authors state the following: “Other techniques, namely, mass spectrometry, immunological methods, genomics and bioinformatics have also have provided critically important information on the previously uncharacterized proteins including miPEPs” or “For example, proteomics is a common technique to reveal huge masses of small uncharacterized proteins in the cellular proteomes”. I was also surprised that the authors did not mention the interactome of miPEP155 and miPEP133 to unravel their protein partners as an example of omics application.
In sum of this part, the authors can add a chapter concerning this and the difficulties in detecting miPEP by untargeted approaches…
Lane 71: correct the sentence to “this method has revealed that some … pri-miRNA fragments can be associated”. Isert ref Lauressergue et al 2022 since TRAP experiements have been done as well.
Lane 81: “For example, proteomics is a common technique to reveal huge masses of small uncharacterized proteins in the cellular proteomes”. Please provide references.
Lane 83: “Other techniques, namely, mass spectrometry, immunological methods, genomics and bioinformatics have also have provided critically important information on the previously uncharacterized proteins including miPEPs”. Please provide references.
Lane 83: remove mass spectrometry since it was already defined earlier by proteomics.
Lane 96: remove microprotein and replace by miPEP. This is a problem all along the ms. Microprotein was defined by S Wenkel as very small protein which correspond to structured domains acting as antagonist of full length proteins. Small ORF peptides are usually defined as SEP and miPEP belongs to this family. In the same vein, micropeptide is inappropriate as a peptide can be defined by 5 amino acids in length. Glutathion is a tripeptide. So, what is a micropeptide? I do agree that this terms can be found in some papers by I think a review must use appropriate nomenclature. Micropeptide and microprotein must be replaced by sORF encoded peptides (generally) and microRNA encoded peptides or miPEPs (for miPEPs).
Lane 136: …primiR156a; ref missing
Lane 143. The level of RNA is measured. So replace transcription by level that is more appropriate.
Lane 166: …gene. ref missing
Lane 205: …transcription activation may relies on a direct interaction…
Lane 272: remove strongly; miPEP8 did not affect strongly…
Lane 290: “ranging in size from several to several dozen”. Rephrase
Lane 313: genes. Ref missing
Lane 316: ref missing
Lane 323: “miPEP-156a is conservative in Brassicaceae plants“. Did the authors meant “conserved”?
Lane 325: “Involvement of the evolutionary conservative At-miPEP164a”. Did the authors meant “conserved”?
Others:
- The figure 1 is of poor quality. Is the original one colored and of high resolution?
- Many pbs concerning the references. The authors have to scrupulously verify them since I did not check all of them.
May be due in part to the ref 42 in refs… shifting numbered ref… But not only…
Lane 237: ref 45 is not the correct one is it ref 46?
Lane 247: is it 48 and not 47?
Lane 250: 50? Not 49 etc…
Lane 276: ref 54, not 53
Lane 278: ref 53 and not 54
Lane 285: ref 53 and not 54
Lane 301: ref 13 and 55 correct? C55 certainly not
Lane 301 ref 53 probably not correct; check. As well 56 and 57.
Lane 312 and 319:not 58 but 59?
Lane 343 not 59 but 60
Lane 346 and 349: not 54 but 53
Lane 355: 61 and not 60
Lane 357: 62?
Lane 358: 53?
Author Response
Comments and Suggestions for Authors
- However, I would suggest to change the title of the review to that of “Genome-wide versus targeted studies of the regulatory miPEPs etc…” . The author’s title implies that the present review will cover Omics studies and how they impacted our knowledge of miPEP’s biology.
- Title has been changed.
- In addition, certain statements of the authors can be misleading. For example, the authors state the following: “Other techniques, namely, mass spectrometry, immunological methods, genomics and bioinformatics have also have provided critically important information on the previously uncharacterized proteins including miPEPs” or “For example, proteomics is a common technique to reveal huge masses of small uncharacterized proteins in the cellular proteomes”.
- These sentences have been removed (see section 1, para 4, lines 4-8).
- Lane 71: correct the sentence to “this method has revealed that some … pri-miRNA fragments can be associated”. Isert ref Lauressergue et al 2022 since TRAP experiements have been done as well.
- Sentence has been changed and the required reference is added (section 1, para 3, last sentence).
- Lane 81: “For example, proteomics is a common technique to reveal huge masses of small uncharacterized proteins in the cellular proteomes”. Please provide references.
- This sentence is removed.
- Lane 83: “Other techniques, namely, mass spectrometry, immunological methods, genomics and bioinformatics have also have provided critically important information on the previously uncharacterized proteins including miPEPs”. Please provide references.
- This sentence is removed.
- Lane 83: remove mass spectrometry since it was already defined earlier by proteomics.
- This sentence is removed.
- Lane 96: remove microprotein and replace by miPEP. This is a problem all along the ms. Microprotein was defined by S Wenkel as very small protein which correspond to structured domains acting as antagonist of full length proteins. Small ORF peptides are usually defined as SEP and miPEP belongs to this family. In the same vein, micropeptide is inappropriate as a peptide can be defined by 5 amino acids in length. Glutathion is a tripeptide. So, what is a micropeptide? I do agree that this terms can be found in some papers by I think a review must use appropriate nomenclature. Micropeptide and microprotein must be replaced by sORF encoded peptides (generally) and microRNA encoded peptides or miPEPs (for miPEPs).
- Words “microprotein” and “micropeptide” have been changed to miPEP and peptide (see section 2, para 1, lines 6, 10, 17; para3, line 2; section 3, para 1, lines 4, 5).
- Lane 136: …primiR156a; ref missing
- Reference is added (Section 2, para 3, line 6).
- Lane 143. The level of RNA is measured. So replace transcription by level that is more appropriate.
- “Transcription” is replaced by “transcription level” (see section 2, para 4, line 4).
- Lane 166: …gene. ref missing
- Reference to “mirbase” is added (section 3, para 1, line 11).
- Lane 205: …transcription activation may relies on a direct interaction…
- The sentence is changed accordingly (see section section 4, para2, line 8).
- Lane 272: remove strongly; miPEP8 did not affect strongly…
- Word “strongly” is removed (see section 5, para 2, line 4).
- Lane 290: “ranging in size from several to several dozen”. Rephrase
- This sentence is rephrased (see section 6, lines 1-4).
- Lane 313: genes. Ref missing
- The references are added (see section 6, para 2, line 5).
- Lane 316: ref missing
- The reference is added (see section 6, para 2, line 11).
- Lane 323: “miPEP-156a is conservative in Brassicaceae plants“. Did the authors meant “conserved”?
- Word “conservative” is replaced by conserved (see section 6, para2, line 19).
- Lane 325: “Involvement of the evolutionary conservative At-miPEP164a”. Did the authors meant “conserved”?
- Word “conservative” is replaced by conserved (see section 6, para2, line 21).
- Many pbs concerning the references. The authors have to scrupulously verify them since I did not check all of them. May be due in part to the ref 42 in refs… shifting numbered ref… But not only…
- Mistaking shift was due to ref. 42, and now all references are in a correct order.